# The Misconception of Antibiotic Equal to an Anti-Inflammatory Drug Promoting Antibiotic Misuse among Chinese University Students

**DOI:** 10.3390/ijerph16030335

**Published:** 2019-01-25

**Authors:** Weiyi Wang, Xiaomin Wang, Yanhong Jessika Hu, Dan Wu, Jingjing Lu, Yannan Xu, Chenhui Sun, Xudong Zhou

**Affiliations:** 1Institute for Social Medicine and Family Medicine, School of Medicine, Zhejiang University, 866 Yuhangtang Road, Hangzhou 310058, China; weiyiwang@zju.edu.cn (W.W.); ellen_wang@zju.edu.cn (X.W.); jingjinglu@zju.edu.cn (J.L.); seanxuzju@zju.edu.cn (Y.X.); sunchenhui@zju.edu.cn (C.S.); 2School of Public Health, The University of Hong Kong, 7 Sassoon Road, Pokfulam 10000, Hong Kong; jesshu17@hku.hk; 3The University of North Carolina at Chapel Hill Project-China, 2 Lujing Road, Guangzhou 510095, China; denisewd@163.com

**Keywords:** misconception, antimicrobial resistance, antibiotic misuse behaviors, university students

## Abstract

Massive misuse of antibiotics is one of the most important reasons for antimicrobial resistance (AMR). Misconceptions of antibiotics contribute to antibiotic misuse behaviors. This study aims to examine whether university students hold the misconception that *Antibiotic is a Xiaoyanyao* (literally means anti-inflammatory drug in Chinese), and association between this misconception and antibiotic misuse behaviors. A cross-sectional study was conducted among university students using the cluster random sampling method in six universities of six regions in China (one university per region). The Chi-square test was used to assess the relationship between the misconception and antibiotic misuse behaviors. Logistic regression was conducted to identify the risk factors for antibiotic misuse behaviors. 11,192 of university students completed the entire questionnaire. There were 3882 (34.7%) students who were considered to have the misconception. Female students were more likely to have the misconception compared with males (36.7% vs. 32.6%, *P* < 0.001). Those students with a background of social science/humanities were more likely to have the misconception compared with those from science and medicine (44.1% vs. 30.3% vs. 20.1%, *P* < 0.001). Students came from rural areas compared with those from urban areas (37.5% vs. 32.5%, *P* < 0.001) were more likely to have the misconception. Students who had the misconception were 1.51 (95% CI 1.21–1.89, *P* < 0.001) times, 1.34 (95% CI 1.21–1.48, *P* < 0.001) times, and 1.36 (95% CI 1.24–1.50, *P* < 0.001) times more likely to report self-medication, request to obtain antibiotics, and take antibiotics prophylactically than those who did not have this misconception, respectively. The high proportion of university students’ misconception on *Antibiotic is a Xiaoyanyao* is worth more attention. Effective health education and interventions need to be promoted among university students and the whole population.

## 1. Introduction

Antimicrobial resistance (AMR) has become one of the greatest threats to global health [1,2,3]. The massive misuse of antibiotics is a leading factor in AMR development [1,4,5,6]. The global consumption of antibiotics increased by 36% from 2000 to 2010 [7]. Studies have shown that poor knowledge regarding appropriate antibiotic use among the general public has led to the misuse of antibiotics [8,9,10]. Misconception of antibiotics is a major form of poor knowledge [11,12]. Since 2015, the World Health Organization (WHO) has defined the third week of November each year as the “World Antibiotics Awareness Week” to increase global awareness and curb the spreading of AMR [13].

In China, people have a long-term misunderstanding of the difference between antibiotics and nonsteroidal anti-inflammatory drugs (NSAIDs, simply called *Xiaoyanyao* in Chinese which literally translates as drugs that can eliminate inflammation). The former works by killing or inhibiting bacteria growth that may help control or cure inflammation caused by bacterial infections [14], while the latter directly reduces inflammatory responses by suppressing the formation of biological chemicals [15]. Previous studies revealed that many Chinese people habitually regard antibiotics as NSAIDs because both can alleviate inflammation [10,16,17]. Such a misperception formed partly due to Chinese doctors’ poor explanations of antibiotics. As bacterial infections can consequently cause inflammations, doctors often simply explain antibiotic as a *Xiaoyanyao* to make it easier for patients to understand the efficacy of antibiotics. Since the general public had limited antibiotic related knowledge, gradually they formed this misperception. Another driving force may be related to inadequate education of health professionals, especially during the era of barefoot doctors who were only trained for three to six months to provide healthcare to rural populations in China [18]. These primary care physicians may not have possessed sufficient knowledge to rationally decide whether a health condition requires antibiotic treatment or not. In addition, antibiotics and *Xiaoyanyao* were interchangeably used by Chinese academics in the 1990s, for example, statements like *“Penicillin - the first choice of Xiaoyanyao in rural areas”* were not uncommon [19]. Therefore, the term *Xiaoyanyao* standing for antibiotics has been widely accepted by the general public in China [10,16,17]. This misconception leads the public to misbelieve that antibiotics can be used to treat simple conditions associated with self-limited inflammation, which may increase the risk of antibiotic misuse. Although recent studies in China have noticed that the misconception *Antibiotic is a Xiaoyanyao* widely exists among the general public and some of the studies have made the misconception one of the items to measure the public’s knowledge level of antibiotics [10,16,17], there is little information about such a misperception and its association with antibiotic misuse behaviors.

As the contemporary elite group, university students represent opinion leaders, and will play an important role in the future trajectory of antibiotic use in China. Our previous studies have shown that university students commonly misused antibiotics [20,21]. This study aims to describe the misconception that *Antibiotic is a Xiaoyanyao* and its association with antibiotic misuse behaviors among Chinese university students.

## 2. Materials and Methods

### 2.1. Study Population and Sampling

A cross-sectional survey was conducted among university students from September to November 2015 in China. China has six geographic regions: north, east, northeast, northwest, central-south, and southwest. From each region, one province was selected, and from each province one comprehensive university was selected, namely Nankai University, Zhejiang University, Jilin University, Lanzhou University, Wuhan University and Guizhou University.

### 2.2. Study Instrument

In this study, a self-administered questionnaire was used. The questionnaire included information about (i) the socio-demographic characteristics of respondents. (ii) antibiotic knowledge, including the item *Antibiotic is a Xiaoyanyao.* (iii) the antibiotic misuse behaviors including self-medication with antibiotics in the past month, which means self-treated with antibiotics without advice from a doctor, nurse, or dentist when ill in the past month, requesting to obtain antibiotics when doctor initially refused to prescribe them in the past year, and taking antibiotics prophylactically in the past year. There are three options in each knowledge item (Yes, No, Do not know). As for the item that *Antibiotic is a Xiaoyanyao*, when respondents chose Yes/Do not know, he/she was considered to have the misconception. The questionnaire was pre-tested among 200 participants. Ethical approval was obtained from Research Ethics Committee of School of Public Health, Zhejiang University.

### 2.3. Data Collection

Data were collected by filling in a web-based questionnaire in Sojump (a Chinese survey tool similar to Survey Monkey). We used cluster random sampling method to select respondents. Students with a background in social science/humanities, sciences, and medicine from each university were sampled. Three investigators were recruited from each study site and were trained before the survey. According to the schedule of each university, classrooms on the day of the survey were randomly selected. The aim of the study was explained to the lecturer and permission was obtained before class. Then all students in the classroom were invited to participate. Students were informed of the purpose of the survey before the printed Quick Response code of the electronic questionnaire was disseminated. Instructions on how to complete the questionnaire were provided. At the beginning of the questionnaire, the anonymity and confidentiality of the study were stressed. More than 95% of students in the selected classes completed the questionnaire, and students who completed the questionnaire were rewarded with 3RMB (0.5 US$) on WeChat via smartphone.

### 2.4. Data Analysis

Data analysis was performed using SPSS version 24.0 (SPSS Inc., Chicago, IL, USA). Descriptive analysis was used to describe the socio-demographic characteristics. The χ^2^ test was used to examine associations between the misconception and antibiotic misuse behaviors. Logistic regression analysis was used to determine factors associated with self-medication with antibiotics (Yes/No), requesting to obtain antibiotics (Yes/No), and taking antibiotics prophylactically (Yes/No). All of socio-demographic characteristics (university, sex, age, educational level, major background, hometown, average monthly household income) were included in the logistic regression model. The significance level (type 1 error rate) was set at 0.05. 

## 3. Results

### 3.1. The Socio-Demographic Characteristics of the Respondents

In total, 11,192 university students completed the questionnaire online and 267 (2.4%) invalid questionnaires were excluded because of missing key variables. Among all, 5515 (49.3%) were male and 5677 (50.7%) were female. The age range of respondents was 16-40, with a mean age of 20.8 (SD = 2.7). Of all the respondents, 4908 (43.9%) with a background of social science/humanities, 4465 (39.9%) from science, and 1819 (16.3%) from medicine. Most students (8892, 79.4%) were undergraduates. More than half of students (6271, 56.0%) came from urban areas, and 3417 (52.0%) had an average monthly household income of no more than 3001-10000 RMB (US$462–1538).

### 3.2. The Misconception Antibiotic is a Xiaoyanyao

Table 1 shows that 3882 (34.7%) students had the misconception *Antibiotic is a Xiaoyanyao*, with the highest proportion in Guizhou University (41.9%) and the lowest in Zhejiang University (22.4%). More females than males (36.7% vs. 32.6%, *P* < 0.001), graduates than undergraduates (37.9% vs. 33.9%, *P* < 0.001) held the misconception. Those who came from rural areas (37.5% vs. 32.5%, *P* < 0.001) and those who had a lower average monthly household income were more likely to have the misconception. Students with a background of social science/humanities were more likely to have the misconception compared with those from science and medicine (44.1% vs. 30.3% vs. 20.1%, *P* < 0.001). 

### 3.3. The Association between the Misconception and Antibiotic Misuse Behaviors

Among 1711 students with self-treatment in the past month, those who misunderstood the conception were more likely to use antibiotics for self-medication (37.0% vs. 25.3%, *P* < 0.001) (Table 2). In the past year before the survey, those with the misconception were more likely to request to obtain antibiotics from doctors (24.5% vs. 17.5%, *P* < 0.001) and take antibiotics for prophylaxis (28.6% vs. 20.0%, *P* < 0.001). As is shown in Table 3, after adjusting for potential confounding factors, students who had the misconception were 1.51 (95% CI 1.21–1.89, *P* < 0.001) times more likely to self-medicate with antibiotics, 1.33 (95% CI 1.21–1.47, *P* < 0.001) times more likely to request to obtain antibiotics, and 1.36 (95% CI 1.24–1.50, *P* < 0.001) times more likely to take antibiotics prophylactically. 

## 4. Discussion

To our knowledge, our study for the first time examined the misconception that *Antibiotic is a Xiaoyanyao* in the Chinese context and its association with antibiotic misuse behaviors among Chinese university students. We found that the misconception was prevalent among the university students and was significantly associated with antibiotic misuse behaviors including self-medication with antibiotics, requesting to obtain antibiotics from doctors, and taking antibiotics prophylactically.

Although WHO and other international organizations have provided the guidelines to combat the increasing AMR globally [4], it is crucial to explore contributing factors of antibiotic misuse and AMR in the local context. Studies have shown that the misconceptions on antibiotics and their appropriate use were common worldwide, such as antibiotics were effective for viral infections [22,23,24]. The misconception *Antibiotic is a Xiaoyanyao* is a unique one in the Chinese context. In the early medical training program in China, doctors often used antibiotics and *Xiaoyanyao* interchangeably, which spread this misconception to patients over time [25].

Our study showed that the misconception held by the general public increased the risk of misuse of antibiotics, including self-medication with antibiotics, requesting antibiotics from doctors, and taking antibiotics prophylactically. This suggests that when people get self-limiting diseases with inflammatory reactions such as sore throat and diarrhea, they may naturally think that they need *Xiaoyanyao* to reduce inflammation, thereby increasing the risk of self-medication with antibiotics. Meanwhile, requests by patients, especially those with flu-like symptoms, have become one of the most important reasons for doctors’ antibiotic prescriptions [26]. Formally educated doctors can distinguish the *Xiaoyanyao* (antibiotics) with anti-inflammatory drugs, however, it is challenging for the public to distinguish without proper interventions. 

According to the results in our study, it is urgent to carry out interventions and health education to correct the misconception of antibiotic equaling to anti-inflammatory drug. From the provider side, terms of *Xiaoyanyao* and antibiotics should be explicitly defined and properly used by providers. They should stop referring to antibiotics in both medical training and patients’ consultation process. From the patients and the general public side, tailored health education is needed to help service users differentiate antibiotics and anti-inflammatory drugs, which might help reduce patients’ expectation of antibiotic use for self-limited conditions. In addition, antibiotic use was informally called Guayanshui (Dadiandi, Guashui), which means using antibiotics intravenously [27]. Posters and brochures printed with the information relating to the distinction between antibiotics and *Xiaoyanyao* can be displayed in medical institutions with high outpatient flow (such as vaccination clinics) to correct the misconception. Furthermore, it is of great importance to distinguish antibiotics by adopting standardized drug name in the local culture. Our previous study showed that Chinese university students had high awareness of antibiotic overuse and its threat to human health, but had too little knowledge to identify specific antibiotics in the market due to various generic and brand names [20,21]. Besides, an appropriate education course can be offered in primary and secondary schools as well as universities to improve students’ knowledge of antibiotics. 

Our results also suggest that health education programs should target certain subpopulations such as female students, those with a social science/humanities background, and students who came from rural areas. In most cases, women are the primary caregivers in a family [28,29,30], which attaches importance to health education being focused on them. Those students from social science/humanities were more likely to misuse antibiotics probably due to less exposure to relevant health information. It is therefore important to raise their awareness. Further, our study focused on young people with relatively high education in the country. We identified a high proportion (over one third) of university students who held the misconception. University students are the next generation of parents of young children, who are known to be very high users of antibiotics. Therefore, educating these young people is crucial to changing the misconception in order to reduce antibiotic misuse in China.

Our study explored the association between the misconception and antibiotic misuse behaviors, which provided evidences for further interventions. We also noted several limitations. As a self-reported questionnaire was used, the accuracy of the information in this study depended on respondents’ responses, which may influenced by social desirability. But this bias has been minimized through voluntary participation and anonymity. Further, university students in the present study were highly educated, thus the results of our study may underestimate the misconception among the general population. Further studies towards the general public regarding the misconception are needed.

## 5. Conclusions

The misconception *Antibiotic is a Xiaoyanyao* among Chinese university students is common. This misconception is significantly associated with self-medication with antibiotics, requesting to obtain antibiotics, and taking antibiotics prophylactically. There is an urgent need to correct this misconception among university students, especially the female students, those with a social science/humanities background, and those who came from rural areas. 

## Figures and Tables

**Table 1 ijerph-16-00335-t001:** The association between socio-demographic characteristics and the misconception (n = 11,192).

Characteristics	Antibiotic Is a *Xiaoyanyao*		χ^2^/t	*P*
	Yes/Do not know [n(%)]	No [n(%)]		
University (province)			220.0	<0.001
Zhejiang University	397 (22.4)	1378 (77.6)		
Guizhou University	851 (41.9)	1179 (58.1)		
Jilin University	777 (39.6)	1184 (60.4)		
Lanzhou University	720 (38.8)	1138 (61.2)		
Nankai University	596 (34.0)	1156 (66.0)		
Wuhan University	541 (29.8)	1275 (70.2)		
Sex			20.83	<0.001
Male	1798 (32.6)	3717 (67.4)		
Female	2084 (36.7)	3593 (63.3)		
Age			9.080	<0.001
Mean (SD)	21.0 (2.7)	20.5 (2.7)		
Education level			12.96	<0.001
Undergraduate	3011 (33.9)	5881 (66.1)		
Graduate	871 (37.9)	1429 (62.1)		
Major background			399.0	<0.001
Social science/humanities	2163 (44.1)	2745 (55.9)		
Science	1353 (30.3)	3112 (69.7)		
Medicine	366 (20.1)	1453 (79.9)		
Home town			30.54	<0.001
Rural	1845 (37.5)	3076 (52.5)		
Urban	2037 (32.5)	4234 (67.5)		
Average monthly household income (RMB)			32.63	<0.001
≤3000 (US$461)	1310 (38.3)	2107 (61.7)		
3001–10,000 (US$462–1538)	1958 (33.6)	3865 (66.4)		
10,001–20,000 (US$1539–3076)	444 (30.9)	991 (69.1)		
>20,000 (US$3076)	170 (32.9)	347 (67.1)		

**Table 2 ijerph-16-00335-t002:** The association between the misconception and antibiotic misuse behaviors by students (n = 11,192).

Behaviors	Antibiotic Is a *Xiaoyanyao*		χ^2^	*P*
	Yes/Do not know [n(%)]	No [n(%)]		
Self-medication with antibiotics ^1^			26.20	<0.001
Yes	233 (37.0)	274 (25.3)		
No	396 (63.0)	808 (74.7)		
Requested to obtain antibiotics			77.02	<0.001
Yes	950 (24.5)	1280 (17.5)		
No	2932 (75.5)	6030 (82.5)		
Took antibiotics prophylactically			107.7	<0.001
Yes	1112 (28.6)	1460 (20.0)		
No	2770 (71.4)	5850 (80.0)		

^1^ 3337 out of 11,192 students had a self-limited disease in the past month before the survey, 1711 of 3337 students had a self-treatment.

**Table 3 ijerph-16-00335-t003:** Logistic regression of the misconception and antibiotic misuse behaviors.

	Self-Medication with Antibiotics (n = 1711) ^1^		Requested to Obtain Antibiotics (n = 11192)		Took Antibiotics Prophylactically (n = 11,192)	
	OR (95 CI)	*P*	OR (95 CI)	*P*	OR (95 CI)	*P*
**Antibiotic is a *Xiaoyanyao***						
No	1		1		1	
Yes/Do not know	1.51 (1.21–1.89)	<0.001	1.34 (1.21–1.48)	<0.001	1.36 (1.24–1.50)	<0.001
**University (province)**						
Zhejiang University	1		1		1	
Guizhou University	2.06 (1.32–3.23)	0.001	1.33 (1.12–1.59)	0.001	1.86 (1.57–2.20)	<0.001
Jilin University	2.84 (1.86–4.34)	<0.001	1.66 (1.40–1.96)	<0.001	1.81 (1.53–2.14)	<0.001
Lanzhou University	2.36 (1.55–3.60)	<0.001	1.37 (1.15–1.63)	<0.001	1.71 (1.45–2.03)	<0.001
Nankai University	2.31 (1.49–3.57)	<0.001	0.84 (0.69–1.01)	0.06	1 (0.83–1.20)	0.969
Wuhan University	1.36 (0.86–2.14)	0.189	0.93 (0.78–1.12)	0.448	1 (0.84–1.19)	0.99
**Sex**						
Male	1		1		1	
Female	1.00 (0.80–1.25)	0.977	1.06 (0.96–1.16)	0.292	1.00 (0.91–1.10)	0.933
**Age**	1.01 (0.97–1.07)	0.582	1.02 (1.00–1.05)	0.054	1.01 (0.99–1.03)	0.518
**Education level**						
Undergraduate	1		1		1	
Graduate	1.12 (0.81–1.55)	0.500	1.12 (0.97–1.30)	0.119	0.90 (0.78–1.03)	0.132
**Major**						
Social science/humanities	1		1		1	
Science	0.94 (0.74–1.21)	0.648	0.75 (0.67–0.83)	<0.001	0.70 (0.63–0.77)	<0.001
Medicine	0.86 (0.63–1.19)	0.364	0.59 (0.51–0.68)	<0.001	0.50 (0.43–0.58)	<0.001
**Home town**						
Rural	1		1		1	
Urban	0.82 (0.64–1.05)	0.112	0.99 (0.89–1.10)	0.798	0.92 (0.83–1.02)	0.129
**Average monthly household income (RMB)**						
≤3000 (US$461)	1		1		1	
3001–10,000 (US$462–1538)	0.97 (0.74–1.27)	0.808	1.04 (0.93–1.17)	0.473	0.92 (0.82–1.03)	0.134
10,001–20,000 (US$1539–3076)	1.19 (0.81–1.76)	0.373	1.06 (0.89–1.27)	0.487	0.91 (0.77–1.08)	0.282
>20,000 (US$3076)	1.03 (0.60–1.77)	0.916	1.01 (0.79–1.30)	0.930	0.81 (0.63–1.03)	0.091

^1^ 3337 out of 11,192 students had a self-limited disease in the past month before the survey, 1711 of 3337 students had a self-treatment.

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
