# Peer review of "The Misconception of Antibiotic Equal to an Anti-Inflammatory Drug Promoting Antibiotic Misuse among Chinese University Students"

_ijerph, 2019, doi:10.3390/ijerph16030335_

Round 1
Reviewer 1 Report
The subject is according to the scope of the Journal and the chosen topic is interesting but, in some pieces of the text, the Authors should devote more attention in presenting the information. I recommend the Authors to improve the Table 1 adding some additional significant information.
Finally, the paper should be revised by a English native speaker in order to remove spelling inaccuracies and to make the manuscript more formal.
Please consider some specific comments and suggestions.
The most meaningful aspect concerning the misconception of antibiotic compared to the anti-inflammatory drug was obtained from the university courses data, rather than sex and demographic characteristics. In fact 44.1% of students belonging to the Social Science/Humanities faculties had misconception about antibiotic compared to Science (30.3%) and Medicine (20.1%) faculties. In that way, we can say that the mainly misconception attitude comes from the students belonging to non scientific courses before focusing on sex or demographic characteristics.
Starting from that information, I would have been interested to deep in the “Major” characteristic.
Then, I would encourage the Authors to detect and illustrate how many female/male and how many students coming from urban/rural areas, with their survey, belonged to the Science and Medicine courses respect Social Science/Humanities courses.
Table 1: What does “t” represent?
Table 3: How did the Authors calculate the logistic regression? Please, include it in Materials and Methods. Moreover, why did the Authors choose the number “1” for some lower data of some study groups and “1” for higher data of other study groups?
I do not agree with the use of “Xiaoyanyao” instead of “nonsteroidal anti-inflammatory drug” or the acronym “NSAIDs” within the manuscript.
That phrases should be reorganized.
26-28: “coming” instead of “came”.
“rural areas (37.5% vs. 32.5%,”) Female students (36.7% vs. 32.6%,). Compared to whom?
“majored in “social science/humanities (P < 0.001),” The percentages are missing.
Moreover, were all the female students coming from rural areas and majored in social science/humanities more likely to have the misconception or are the Authors referring to 3 separated groups?
28-30: “the students who have the misconception were 1.51…. more likely to engage in self-medication…” . Compared to whom?
Abstract: I would avoid the use of “;” when it is not necessary.
I would suggest to the Authors to use the article “the” and conjunction “that” in many pieces of the text.
I would suggest expressions like “the request to obtain antibiotics” instead of “asking for antibiotics”
49-50: This sentence is not understandable. Could the Author rephrase it?
53: What is the argument that “title” is referring to?
58-60: This phrase should be reorganized. The style makes the meaning of this sentence unclear.
74-79: This phrase is not well explained and clear and should be reorganized. Please, could the Authors be more accurate and perceptible?
94: What SPSS is?
101-102 and other pieces of text: How is it possible that all the students of the study were majored? All the students belong to Faculties but not all were majored.
108: Did the Authors refer to “undergraduates” instead of “postgraduates”?
144: Those information are not necessary: “sore throat (Houlongfayan in Chinese, inflammation in the throat) and diarrhea (Changdaofayan in Chinese inflammation in the intestinal tract), ”
147: “distinguish the Xiaoyanyao (antibiotics) with anti-inflammatory drugs”. Are Xiaoyanyao antibiotics or anti-inflammatory drugs as defined at the beginning of the manuscript?
148: I do not agree with the use of “to do so”. Please, correct the style.
Author Response
Dear reviewer,
Thank you so much for your precious comments! We have done some addition analysis and have changed accordingly. Please kindly find our detailed responses followed.
Best regards,
Xudong
Point 1:The subject is according to the scope of the Journal and the chosen topic is interesting but, in some pieces of the text, the Authors should devote more attention in presenting the information. I recommend the Authors to improve the Table 1 adding some additional significant information.
Response 1:Thank you. We have changed accordingly and added some sentences below:
Results part: Students with a background ofsocial science/humanities were more likely to have the misconception compared with those from science and medicine(44.1% vs. 30.3% vs. 20.1%, P < 0.001). (Line 135-138)
Discussion part: Those students from social science/humanities were more likely to misuse antibiotics probably due to less exposure to relevant health information. It is therefore important to raise their awareness. (Line 203-205).
Point 2: Finally, the paper should be revised by a English native speaker in order to remove spelling inaccuracies and to make the manuscript more formal.
Response 2: Thank you. We have invited an English native speaker with background in public health to proofread and improve the manuscript.
Please consider some specific comments and suggestions.
Point 3: The most meaningful aspect concerning the misconception of antibiotic compared to the anti-inflammatory drug was obtained from the university courses data, rather than sex and demographic characteristics. In fact 44.1% of students belonging to the Social Science/Humanities faculties had misconception about antibiotic compared to Science (30.3%) and Medicine (20.1%) faculties. In that way, we can say that the mainly misconception attitude comes from the students belonging to non scientific courses before focusing on sex or demographic characteristics.
Starting from that information, I would have been interested to deep in the “Major” characteristic.Then, I would encourage the Authors to detect and illustrate how many female/male and how many students coming from urban/rural areas, with their survey, belonged to the Science and Medicine courses respect Social Science/Humanities courses.
Response 3: Yes, we agree with your helpful suggestions. Students with different background performed differently of the misconception about antibiotics. Nevertheless, as an early exploratory study, we believe it is equally important to explore which demographic characteristics are correlates of such a misperception because these may also inform and tailor future interventions for specific subgroups. We have done a logistic regression analysis about misconception and characteristics as below. Table shows that sex, age, major and residency are all significant to misconception. Thus, major as well as other demographic characteristics should been given equal consideration as they all have statistical significant in influencing student’s misconception.
Logistic regression of association between misconception and characteristics
Characteristics | Antibiotic is a Xiaoyanyao | |
OR(95CI) | P | |
University(province) | ||
Zhejiang University | 1 | |
Guizhou University | 1.04(0.91-1.19) | 0.576 |
Jilin University | 0.97(0.85-1.11) | 0.664 |
Lanzhou University | 0.73(0.64-0.84) | <0.001 |
Nankai University | 0.66(0.57-0.76) | <0.001 |
Wuhan University | 0.53(0.45-0.61) | <0.001 |
Sex | ||
Male | 1 | |
Female | 1.14(1.05-1.24) | 0.002 |
Age | 1.07(1.05-1.09) | <0.001 |
Education level | ||
Undergraduate | 1 | |
Graduate | 0.96(0.84-1.09) | 0.511 |
Major | ||
Social science/humanities | 1 | |
Science | 0.61(0.55-0.67) | <0.001 |
Medicine | 0.30(0.26-0.34) | <0.001 |
Home town | ||
Rural | 1 | |
Urban | 0.87(0.80-0.96) | 0.004 |
Average monthly household income (RMB) | ||
≤3000 (US$461) | 1 | |
3001–10,000 (US$462–1538) | 0.90(0.81-0.99) | 0.036 |
10,001–20,000 (US$1539–3076) | 0.84(0.73-0.98) | 0.025 |
>20,000 (US$3076) | 0.93(0.75-1.15) | 0.516 |
Point 4:Table 1: What does “t” represent?
Response 4: The “t” here represents “t-test”. In characteristics, Age is a continuous variable, so t-test was used to compare two groups.
Point 5:Table 3: How did the Authors calculate the logistic regression? Please, include it in Materials and Methods. Moreover, why did the Authors choose the number “1” for some lower data of some study groups and “1” for higher data of other study groups?
Response 5: The calculation of logistic regression has been added to Materials and Methods. Please refer to the manuscript part of materials and methods:
Logistic regression analysis was used to determine factors associated with self-medicationwith antibiotics (Yes/No), requesting to obtainantibiotics (Yes/No), and taking antibiotics prophylactically (Yes/No).All of socio-demographic characteristics (university, sex, age, educational level, major background, hometown, average monthly household income) were included in the logistic regression model. The significance level (type 1 error rate) is set at 0.05.(Line 116-120)
And in table 3, “1” represents the reference group, the first variable was selected as the reference group of each categorical variable in logistic regression, and the order of theses variables is consistent with Table 1. Thus, this may occur the situation that “1” for some lower data of some study groups and “1” for higher data of other study groups.
Point 6: I do not agree with the use of “Xiaoyanyao” instead of “nonsteroidal anti-inflammatory drug” or the acronym “NSAIDs”within the manuscript.
That phrases should be reorganized.
Response 6:In China, “nonsteroidal anti-inflammatory drug” are commonly referred to Xiaoyanyao, and antibiotics are commonly referred to one kind of Xiaoyanyao. We use this phrase to explain that the misconception is a unique one in the Chinese context and attach importance to it. The way of using local language to explain the misconception of antibiotic among the public was used by researchers in other countries such as Thailand researchers [1].
References:
[1] Sumpradit, N.; Chongtrakul, P.; Anuwong, K.; Pumtong, S.; Kongsomboon, K.; Butdeemee, P.; Khonglormyati, J.; Chomyong, S.; Tongyoung, P.; Losiriwat, S.; Seesuk, P.; Suwanwaree, P.; Tangcharoensathien, V., Antibiotics Smart Use: a workable model for promoting the rational use of medicines in Thailand. Bull World Health Organ2012,90, 905-913.
Point 7: 26-28: “coming” instead of “came”.
Response 7: We have revised and completed the sentences in the revised manuscript as below.
Students coming from rural areas compared with those from urban areas (37.5% vs. 32.5%, P < 0.001) were more likely to have the misconception. (Line 31)
Point 8: “rural areas (37.5% vs. 32.5%,”) Female students (36.7% vs. 32.6%,). Compared to whom? “majored in “social science/humanities (P < 0.001),” The percentages are missing.
Response 8:We have added the missing data and completed the sentences in the revised manuscript as below.
Female students were more likely to have the misconception compared with male (36.7% vs. 32.6%, P < 0.001). Those students with a background of social science/humanities were more likely to have the misconception compared with those from science and medicine (44.1% vs. 30.3% vs. 20.1%, P < 0.001). (Line 28-31)
Point 9: Moreover, were all the female students coming from rural areas and majored in social science/humanities more likely to have the misconception or are the Authors referring to 3 separated groups?
Response 9: We agree that was unclear. This point refers to 3 separated groups. We have
revised the sentences and please refer to the revised version of the manuscript as below.
Female students were more likely to have the misconception compared with male (36.7% vs. 32.6%, P < 0.001). Those students with a background of social science/humanities were more likely to have the misconception compared with those from science and medicine (44.1% vs. 30.3% vs. 20.1%, P < 0.001). Students comingfrom rural areas compared with those from urban areas (37.5% vs. 32.5%, P < 0.001) were more likely to have the misconception.(Line 28-33)
Point 10: 28-30: “the students who have the misconception were 1.51…. more likely to engage in self-medication…”. Compared to whom?
Response 10: Students who had the misconception were 1.51 (95% CI 1.21–1.89, P < 0.001) times, 1.34 (95% CI 1.21–1.48, P < 0.001) times, and 1.36 (95% CI 1.24–1.50, P < 0.001) times more likely to report self-medication, request to obtain antibiotics, and take antibiotics prophylactically than those who did not have this misconception, respectively.(Line 33-36)
Point 11: Abstract: I would avoid the use of “;” when it is not necessary.
Response 11:We have deleted those unnecessary “;” in the revised manuscript.
Point 12: I would suggest to the Authors to use the article “the” and conjunction “that” in many pieces of the text.
Response 12: We have revised the article with “the” and conjunction “that” where necessary.
Point 13: I would suggest expressions like “the request to obtain antibiotics” instead of “asking for antibiotics”
Response 13: We have changed the expressions in the revised manuscript.
Point 14: 49-50: This sentence is not understandable. Could the Author rephrase it?
Response 14:We have rephrased this sentence in the revised manuscript.
As bacterial infections can consequently cause inflammations, doctors often simply explain antibioticas a Xiaoyanyaoto make it easier for patients to understand the efficacy of antibiotics.Since the general public had limited antibiotic related knowledge, gradually they formed this misperception.(Line 57-59)
Point 15: 53: What is the argument that “title” is referring to?
Response 15: We have rephrased this sentence. We were not able to find any relevant English references here, but there were related Chinese references which showed that Antibiotics and Xiaoyanyao were interchangeably used by Chinese academics expressions [1]. You can find this Chinese literature at the following link:
http://www.cqvip.com/qk/80701x/199402/1005108858.html
References:
[1] Zhou, X. Penicillin - the first choice of Xiaoyanyao in rural areas. Xin Nong Cun1994, 2, 32 (in Chinese)
Point 16: 58-60: This phrase should be reorganized. The style makes the meaning of this sentence unclear.
Response 16:We have reorganized this sentenced in the revised manuscript to make it more clearly.
Although recent studies in China have noticed that the misconception Antibiotic is a Xiaoyanyaois widely existed among the general public and some of the studies have made the misconception as one of the items to measurethe public’sknowledge level ofantibiotic [10, 16, 17], there is little information about such a misperception and its association with antibiotic misuse behaviors. (Line 70-74)
Point 17: 74-79: This phrase is not well explained and clear and should be reorganized. Please, could the Authors be more accurate and perceptible?
Response 17:We have reorganized this part in the revised manuscript to make it more clearly.
In this study, a self-administered questionnaire was used. The questionnaire included information about (i) the socio-demographic characteristics of respondents. (ii) antibiotic knowledge, including the item Antibiotic is a Xiaoyanyao.(iii)the antibiotic misuse behaviors including self-medication with antibiotics in the past month, which means self-treated with antibiotics without advice from a doctors, nurse, or dentist when ill in the past month, requesting to obtain antibiotics when doctor initially refused to prescribe them in the past year, and taking antibiotics prophylactically in the past year. There are three options in each knowledge item (Yes, No, Do not know). As for the item that Antibiotic is a Xiaoyanyao, when respondents chose Yes/Do not know, he/she was considered to have the misconception. (Line 87-96)
Point 18: 94: What SPSS is?
Response 18: SPSS (Statistic Package for Social Science) is a powerful tool that is capable of conducting nearly any type of data analysis used in the social sciences [1]. We have added the following sentence in the manuscript: Data analysis in this study was performed using SPSS version 24.0 (SPSS Inc., Chicago, IL, USA).
References:
[1]Mallery, P.; George, D. SPSS for Windows Step-by-Step, 1st ed.; World Publishing Corporation: Beijing, China, 2006.
Point 19: 101-102 and other pieces of text: How is it possible that all the students of the study were majored? All the students belong to Faculties but not all were majored.
Response 19: All of students majored in different kinds of majors, and those majors can be divided into Social science/humanities, Science and Medicine. We have rephrased a statement about this point using “with the background of” instead of “majored in”.
Point 20: 108: Did the Authors refer to “undergraduates” instead of “postgraduates”?
Response 20:Yes, thanks for spotting the typo. We have revised in the manuscript.
Point 21: 144: Those information are not necessary: “sore throat (Houlongfayan in Chinese, inflammation in the throat) and diarrhea (Changdaofayan in Chinese inflammation in the intestinal tract), ”
Response 21:We have deleted these information in the revise manuscript.
Point 22: 147: “distinguish the Xiaoyanyao (antibiotics) with anti-inflammatory drugs”. Are Xiaoyanyao antibiotics or anti-inflammatory drugs as defined at the beginning of the manuscript?
Response 22: Yes, we have defined these two different drugs at the beginning of the article, in the section of introduction.
In China, people have a long-term misunderstanding of the difference between antibiotics and nonsteroidal anti-inflammatory drugs (NSAIDs, simply called Xiaoyanyaoin Chinese which literally translates as drugs that can eliminate inflammation). The former works by killing or inhibiting bacteria growth that may help control or cure inflammation caused by bacterial infections [14], while the latter directly reduces inflammatory responses by suppressing the formation of biological chemicals [15].(Line 50-54)
Point 23: 148: I do not agree with the use of “to do so”. Please, correct the style.
Response 23:We have rephrased and corrected the style of this sentence in the revised manuscript.
however, it is challenging for the public to distinguish without proper interventions. (Line 179)

Reviewer 2 Report
The article entitled "The misconception of antibiotic equal to an anti-inflammatory drug promoting antibiotic misuse among Chinese university students" by Wang et al., has been designed properly but the representation of the data should be improved for better understanding.
I would also like to stress that the manuscript should be checked with a native English speaker as throughout the manuscript the sentences are not clear enough to understand for wide reader.
I have some minor point should be kept in the consideration before publication.
Line 19: rephrase the sentence
Line 23-24: rephrase the sentence.
Line 25: does total students with misconception incluse male and felame students both.
Line 26: Female students (36.5% vs 32.6%, P>0.001)- What does it mean? Authors are comparing the percentage with what...Please clarify?
Line 32: Why every time Antibiotic appears with capital A and not antibiotic. Authors are also requested to abbreviate the word "Antibiotic is Xiaoyanyao" as every time it appears will confuse the reader with grammatical error.
Section Introduction: Authors are requested to introduce the subject more generally and broadly that how and why Xiaoyanyao is prescribed by the doctors and other reports on its misconception.
It is also advised to author to divide the sampling population on the basis of age group or at least mention the age range.
Line 75 and Line 76: Check for typo errors. And also from other place throughout the manuscript.
In my personal point of view the questionnaire for the students are very important and should be provided as supplementary datasheet.
It is also important to understand and know the question asked among the university students and medicine students.
Which year of medicine students were included or surveyed as this also plays a vital role for there misconception.
It is also advised to include some to the regular medicine practitioner in the study to understand misconceptions among them and to know if its general misconception came from culture or from the study point of view.
Author Response
Dear reviewer,
Thank you so much for your precious comments! We have done some addition analysis and have changed accordingly. Please kindly find our detailed responses followed.
Best regards,
Xudong
Point 1: The article entitled "The misconception of antibiotic equal to an anti-inflammatory drug promoting antibiotic misuse among Chinese university students" by Wang et al., has been designed properly but the representation of the data should be improved for better understanding.
Response 1: Thank you. We have changed accordingly.
Point 2: I would also like to stress that the manuscript should be checked with a native English speaker as throughout the manuscript the sentences are not clear enough to understand for wide reader.
Response 2: We have invited an English native speaker with background in public health to proofread and improve the manuscript.
Point 3: Line 19: rephrase the sentence
Response 3: We have rephrased the sentence in the revised manuscript.
This study aims to examine whether university students hold the misconception that Antibiotic is a Xiaoyanyao(literally means anti-inflammatory drug in Chinese), and association between this misconception and antibiotic misuse behaviors. (Line 19-22)
Point 4:Line 23-24: rephrase the sentence.
Response 4: We have rephrased the sentence in the revised manuscript.
Chi-squaretestwas used to assess the relationship between the misconception and antibiotic misuse behaviors. (Line 24-25)
Point 5:Line 25: does total students with misconception incluse male and female students both.
Response 5: Yes, in total, there were 3882 (34.7%) students who were considered having the misconception, including1798 male students (32.6%) and 2084 female students (36.7%).
Point 6:Line 26: Female students (36.5% vs 32.6%, P>0.001)- What does it mean? Authors are comparing the percentage with what...Please clarify?
Response 6:We have added the missing data and completed the whole the sentences in the revised manuscript as below.
Female students were more likely to have the misconception compared with male (36.7% vs. 32.6%, P < 0.001). Those students with a background of social science/humanities were more likely to have the misconception compared with those from science and medicine (44.1% vs. 30.3% vs. 20.1%, P < 0.001). Students comingfrom rural areas compared with those from urban areas (37.5% vs. 32.5%, P < 0.001) were more likely to have the misconception.(Line 28-33)
Point 7: Line 32: Why every time Antibiotic appears with capital A and not antibiotic. Authors are also requested to abbreviate the word "Antibiotic is Xiaoyanyao" as every time it appears will confuse the reader with grammatical error.
Response 7: We totally agree with you and we have revised and used the italic format to indicate that this misconception is a specific statement, with capital A only when we describe the misconception.
Point 8: Section Introduction: Authors are requested to introduce the subject more generally and broadly that how and why Xiaoyanyao is prescribed by the doctors and other reports on its misconception.
Response 8: As bacterial infections can consequently cause inflammations, doctors often simply explain antibioticas a Xiaoyanyaoto make it easier for patients to understand the efficacy of antibiotics.Since the general public had limited antibiotic related knowledge, gradually they formed this misperception. Another driving force may be related to inadequate education of health professionals, especially during the era of barefoot doctors who were trained for three to six months only to provide healthcare to rural population in China [18]. These primary care physicians may not have sufficient knowledge to rationally decide whether a health condition requires antibiotic treatment or not. (Line 57-63)
Point 9: It is also advised to author to divide the sampling population on the basis of age group or at least mention the age range.
Response 9:We have added the age range in the revised manuscript as below:
The age range of respondents was 16 to 40, with a mean age of 20.8 (SD=2.7). (Line 126)
Point 10:Line 75 and Line 76: Check for typo errors. And also from other place throughout the manuscript.
Response 10: Thank you very much for spotting the typos. We have corrected the errors and reorganized the sentences in the revised manuscript.
Point 11: In my personal point of view the questionnaire for the students are very important and should be provided as supplementary datasheet.
Response 11:We haveaddedthe questionnaire as supplementary materials to the journal.
Point 12: It is also important to understand and know the question asked among the university students and medicine students.
Response 12:This study used the same questionnaire for all respondents in different academic backgrounds. We have uploaded the questionnaire as supplementary material.
Point 13: Which year of medicine students were included or surveyed as this also plays a vital role for this misconception.
Response 13: We agree that which year of medicine students played a vital role for this misconception. Please refer to our previous paper regarding more details about medical students [1]. In our study, medical students in grade 1 to 8 were all included in the survey. In China, only medical students in senior grades have clinical rotations. We therefore divided them into pre-clinical (Uy1–Uy3) and clinical medical students (Uy4–Uy8) to analyse the difference. To avoid the potential duplication, we did not include year of medicine students in this manuscript.
References:
[1] Hu, Y.; Wang, X.; Tucker, J.D.; Little, P.; Moore, M.; Fukuda, K.; Zhou, X. Knowledge, Attitude, and Practice with Respect to Antibiotic Use among Chinese Medical Students: A Multicentre Cross-Sectional Study. Int. J. Environ. Res. Public Health2018, 15, 1165.
Point 14: It is also advised to include some to the regular medicine practitioner in the study to understand misconceptions among them and to know if its general misconception came from culture or from the study point of view.
Response 14:We totally agree with you that it will be better if we include some regular medicine practitioner in the study to understand misconceptions among them. As we stated in the limitation section (Line 214-216), the target population of our study are university students, and the results may underestimate the misconception. We agree that further studies towards the general public including regular medicine practitioners regarding the misconception are needed.

Reviewer 3 Report
Tackling against antimicrobial resistance is integral to public health. Antimicrobial resistance is related to antimicrobial consumption and every country had differences in misuse behaviours related to antibiotics.
Methods:
I would need to understand several antibiotic misuse behaviors:
"Self-medication in the last month".
Is it possible to get an antibiotic without a medical prescription? In Spain, it is easy for anti-inflammatory drugs but it is more difficult to get an antibiotic because you must go to visit a general practitioner.
Results: 1711 students had self-treatment with antibiotics. How students get self-medication with antibiotics?
In what kind of prophylaxis did students use antibiotics?
Discussion:
Correlation with data about use of antibiotics and antibiotic resistance in the 6 provinces selected would be very interesting.
It might be worth adding more sentences about how to change this misconception among people in China in order to get a better use of antibiotics.
From an ethical point of view:
Were students involved in the recruitment of the study? Or only teachers/lecturer?
It had better completed the questionaire out of classroom to ensure voluntary participation and anonymity.
How will the results be dissseminated to study participants? I think, this is very important to correct this misconception among them.
Author Response
Dear reviewer,
Thank you so much for your precious comments! We have done some addition analysis and have changed accordingly. Please kindly find our detailed responses followed.
Best regards,
Xudong
Tackling against antimicrobial resistance is integral to public health. Antimicrobial resistance is related to antimicrobial consumption and every country had differences in misuse behaviours related to antibiotics.
Methods:
Point 1: I would need to understand several antibiotic misuse behaviours: "Self-medication in the last month".
Response 1:Self-medication with antibiotics in the last month means self-treated with antibiotics without advice from a doctors, nurse, or dentist when got sick in the past month. It is how we stated in the questionnaire, before the self-medication questions. We have added the above definition in the manuscript.
Point 2: Is it possible to get an antibiotic without a medical prescription? In Spain, it is easy for anti-inflammatory drugs but it is more difficult to get an antibiotic because you must go to visit a general practitioner.
Response 2: Yes, it is possible to get an antibiotic without a medical prescription in China [1, 2]. Despite the fact that sales of antibiotics have been restricted to prescription-only by the China Food and Drug Administration since 2004 [3], the policy implementation has been poor.
References
[1] Fang, Y. China should curb non-prescription use of antibiotics in the community. BMJ. 2014, 348, g4233.
[2] Kan, Q.; Wen, J.; Liu X.; Li, Zhen. Inappropriate use of antibiotics in children in China. Lancet 2016, 387, 1273–1274.
[3] State Food and Drug Administration. The five prescription sales only antibacterial released by SFDA. 2004. Available online: http://eng.sfda.gov.cn/WS03/CL0757/61674.html(assessed on 21 December 2018)
Results:
Point 3:1711 students had self-treatment with antibiotics. How students get self-medication with antibiotics?
Response 3:Students get self-medication when they reported that they had experienced a self-limiting illness in the past month, including common cold, sore throat, diarrhea, fever and headache, sometimes with some obvious overlap between symptoms. They treated themselves with antibiotics, which they purchased from a drug store with or without prescriptions, leftover from previous consultation with a doctor, or even given by others, these antibiotics included penicillin, cephalosporins, macrolides, quinolones.
Point 4:In what kind of prophylaxis did students use antibiotics?
Response 4:The most common type of prophylaxis here is that when a mild cold-like symptom appears, would these students use antibiotics for the prevention of common cold or not.
Discussion:
Point 5:Correlation with data about use of antibiotics and antibiotic resistance in the 6 provinces selected would be very interesting.
Response 5: We agree. However, it is a shame that we did not test the antibiotic resistance in the 6 provinces. Moreover, our target population in this survey were university students who came from different regions of the whole country, rather than general population. Therefore, it would be less desirable for us to discuss the antimicrobial resistance in the 6 provinces in the current manuscript. However, our unpublished data cover other populations including migrant workers, young parents, etc. which might allow us to discuss the use of antibiotics and antimicrobial resistance in the future.
Point 6:It might be worth adding more sentences about how to change this misconception among people in China in order to get a better use of antibiotics.
Response 6: We have added some sentences below in the discussion section as below:
Besides, an appropriate education course can be offered in primary and secondary schools as well as universities to improve students’ knowledge of antibiotics. (Line 196)
Posters and brochures printed with the information relating to the distinction between antibiotic and Xiaoyanyaocan be displayed in medical institutions with high outpatient flow (such as vaccination clinics) to correct the misconception. (Line 190-192)
From an ethical point of view:
Point 7:Were students involved in the recruitment of the study? Or only teachers/lecturer?
Response 7:Mainly students were involved in the recruitment of the study. At each university, three investigators approached teachers, explained the aim of the survey and asked for permission to speak to students before the class began. The investigator then explained the aim of the survey to the students, disseminated the printed QR code of the electronic questionnaire, and explained to students how to complete the electronic questionnaire. The first section of the questionnaire consisted of an information sheet and consent form. that was signed-off by all participants. We explained clearly that participation was fully voluntary and anonymous. The questionnaire would take around 5 min to complete.Ethical approval was obtained from Research Ethics Committee of School of Public Health, Zhejiang University.
Point 8:It had better completed the questionaire out of classroom to ensure voluntary participation and anonymity.
Response 8: Yes, we agree. It would be better to ensure the voluntary participation to conducted the survey out of classroom. The choice of conduct the survey in the classroom is to ensure the random sampling approach. However, we paid high attention to the ethical appropriateness during every steps of the study. Only students were recruiting respondents after obtaining teachers’ permission. It was explained clearly to every student that the participation was not compulsory and the anonymity were well guaranteed. We used an electronic questionnaire and no other private information were collected such as name.
Point 9:How will the results be disseminated to study participants? I think, this is very important to correct this misconception among them.
Response 9:We totally agree with you. We have conducted a national campaign which lasted for seven months after this study. One of the most important part was that we have launched and organized a national crowdsourcing contest in university students to solicit submissions on how to reduce antimicrobial resistance (AMR) and antibiotic misuse. All 6 investigated universities were involved in the competition. During the contest promotion, we have also added relevant educational information based on the study results, aiming to promote awareness and reduce the misconception among university students.

Round 2
Reviewer 1 Report
Dear Authors,
you have correctly amended and updated the manuscript to a suitably clarified and proper version.
Reviewer 2 Report
The manuscript entitled "The misconception of antibiotic equal to an anti-inflammatory drug promoting antibiotic misuse among Chinese university students" has been improved after the careful revision by Wang et al and hence can be accepted for publication.
Reviewer 3 Report
I accept the new version and corrections to manuscript.